# Sample Entropy Improves Assessment of Postural Control in Early-Stage Multiple Sclerosis

**DOI:** 10.3390/s24030872

**Published:** 2024-01-29

**Authors:** L. Eduardo Cofré Lizama, Xiangyu He, Tomas Kalincik, Mary P. Galea, Maya G. Panisset

**Affiliations:** 1Department of Medicine, The University of Melbourne, Melbourne, VIC 3052, Australia; xiangyuh1@student.unimelb.edu.au (X.H.); m.galea@unimelb.edu.au (M.P.G.); mpanisset@unimelb.edu.au (M.G.P.); 2Neuroimmunology Centre, Department of Neurology, Royal Melbourne Hospital, Melbourne, VIC 3052, Australia; tomas.kalincik@unimelb.edu.au; 3Clinical Outcomes Research Unit, The University of Melbourne, Melbourne, VIC 3052, Australia; 4Department of Rehabilitation, Royal Melbourne Hospital, Melbourne, VIC 3052, Australia; 5Australian Rehabilitation Research Centre, Royal Melbourne Hospital, Melbourne, VIC 3052, Australia

**Keywords:** stability, posturography, balance, sample entropy, regularity, jerk

## Abstract

Postural impairment in people with multiple sclerosis (pwMS) is an early indicator of disease progression. Common measures of disease assessment are not sensitive to early-stage MS. Sample entropy (SE) may better identify early impairments. We compared the sensitivity and specificity of SE with linear measurements, differentiating pwMS (EDSS 0–4) from healthy controls (HC). 58 pwMS (EDSS ≤ 4) and 23 HC performed quiet standing tasks, combining a hard or foam surface with eyes open or eyes closed as a condition. Sway was recorded at the sternum and lumbar spine. Linear measures, mediolateral acceleration range with eyes open, mediolateral jerk with eyes closed, and SE in the anteroposterior and mediolateral directions were calculated. A multivariate ANOVA and AUC-ROC were used to determine between-groups differences and discriminative ability, respectively. Mild MS (EDSS ≤ 2.0) discriminability was secondarily assessed. Significantly lower SE was observed under most conditions in pwMS compared to HC, except for lumbar and sternum SE when on a hard surface with eyes closed and in the anteroposterior direction, which also offered the strongest discriminability (AUC = 0.747), even for mild MS. Overall, between-groups differences were task-dependent, and SE (anteroposterior, hard surface, eyes closed) was the best pwMS classifier. SE may prove a useful tool to detect subtle MS progression and intervention effectiveness.

## 1. Introduction

Multiple sclerosis (MS) is a neurodegenerative disease that causes damage to the myelinated sheath around nerve axons in the central nervous system (brain, spinal cord, and optic nerve), impairing transmission of action potentials along the axons. This causes a range of sensorimotor symptoms including balance impairment, which tends to worsen with disease progression, and is associated with later gait disturbance and falls [1]. The disability level of patients with MS is commonly evaluated by the Expanded Disability Status Scale (EDSS). The EDSS is based on clinical examination of multiple functional systems [2]. A higher EDSS score indicates a higher level of disability, with a score of 0 meaning no disability and a score of 10 representing death due to MS [3]. Because clinical management of MS relies on early initiation of disease-modifying therapeutics (DMTs) to minimize long-term disability, tools to identify early disease activity are needed. People with MS (pwMS) who have an EDSS score < 4 (pwMS^0–4^) have no limitations in their walking distance; however, changes in postural stability have been reported even in people with EDSS < 3 [1,4,5]. Thus, deterioration in postural stability may be an early indicator of MS disease progression.

Posturography has been widely used to quantify balance control during upright stance under different sensory conditions, including in pwMS [5]. The most common posturographic method uses force plates to obtain centre of pressure (CoP) displacement measures, e.g., velocity [6]. However, the affordability and portability of inertial measurement sensors (IMUs) and their ability to obtain posturographic measurements has led to their rapid assimilation in clinical settings [6,7]. Posturographic measures obtained with IMUs have been shown to be significantly correlated (ICC [0.70–0.94]) to “gold-standard” force plate measures [6]. Although wearable sensors are becoming the preferred option to monitor balance issues in people with neurological diseases [5], it is yet to be determined which measures are more sensitive to impairment, disease progression, and/or interventions.

Linear measures of postural control are most commonly used to quantify balance impairments with IMUs in clinical populations. Although IMU-based systems can provide a broad range of linear measures, few of them have shown to be able to discriminate or differentiate between healthy controls (HC) and pwMS with mild-to-low disability, namely related to range of acceleration [8] and jerk [9]. For example, Spain et al. (2012) found reduced jerk in the mediolateral direction (ML) during an eyes-closed (EC) on a hard surface (H) condition (Jerk_EC-H-ML_) as a significant indicator of MS disease (median EDSS = 3.0, range 0–5) [9]. Solomon et al. (2015), on the other hand, reported that the range of sway acceleration in the mediolateral (ML) direction during an eyes-open (EO) when standing on a foam (F) surface (AR_EO-F-ML_) was one of the strongest independent factors to classify pwMS (median EDSS = 2, range 1.0–2.5) and HC [8].

More recently, non-linear measures of postural stability have shown a greater ability to characterise pathological behaviour [10]. Non-linear measures quantify the dynamic behaviour of human movement by looking at the time-varying structure of, e.g., acceleration signals. One of the most common non-linear measures used in posturographic studies is sample entropy (SE) [7]. Higher values of SE are thought to indicate greater variability of postural responses, affording a greater repertoire of motor strategies to handle perturbations. On the contrary, a lower value may indicate a restriction to more stereotyped motor strategies, which can limit the ability to deal with perturbations occurring in daily life [7]. SE reduction has also been observed in pwMS who exhibit lower limb muscle fatigue [11]. Using force plates, Sun et al. identified SE_EO-H-AP_ as the strongest single classifier of pwMS with a low risk of falls, performing above clinical balance tests (e.g., Berg Balance Score) and standard linear force plate measures (e.g., sway area). However, as the feature importance of SE_EO-H-AP_ was only 15.3%, the authors suggested that more complex postural tasks may provide better discriminative ability [12]. Further to this suggestion, Carpinella et al. (2022) found that pwMS exhibited lower SE than HC during an eyes closed, standing on foam condition [13].

It is noteworthy that the majority of previous studies have focused on measuring postural control, linear and non-linear, in pwMS who have developed evident motor control symptoms [6,14]. However, a more recent study suggests that a complexity index that includes SE**_EC-F_** has an acceptable discriminant ability in pwMS at early stages of the disease (EDSS ≤ 2.5) [13]. Despite this evidence, it is still not known whether non-linear posturography measures, e.g., SE, may better identify subtle impairments of motor control in pwMS with milder symptoms than currently identified linear measures (e.g., jerk and range of acceleration). The latter may help identify novel biomarkers of MS progression at the early stages of the disease, which could facilitate more timely interventions [8].

The aims of this study were to compare the postural performance of pwMS and HC and examine the sensitivity to MS of SE and linear measures. A secondary aim was to assess the discriminability of the same measures to mild MS (EDSS ≤ 2.0; pwMS^0–2^). We hypothesised that (a) SE measures would be significantly lower in pwMS compared to HC, and that (b) SE measures would have higher sensitivity and specificity to MS than linear measures. For our secondary aim, we hypothesised that SE would be better able to classify people with mild MS (EDSS ≤ 2.0).

## 2. Materials and Methods

***Participants:*** 58 people with relapsing-remitting multiple sclerosis (pwMS) and 23 healthy adults provided written informed consent. Inclusion criteria for pwMS included (a) <15 years since onset, and (b) EDSS < 4.0. Exclusion criteria for both groups included the presence of (a) other neurological conditions, (b) cardiovascular disease, (c) orthopaedic conditions, and (d) pregnancy or being <5 months post-partum.

***Protocol:*** Postural control was measured using two Opal IMUs (APDM, Portland, OR, USA) placed at the sternum and lumbar spine. We selected these locations as previous studies have derived pwMS classifier metrics from both sites using the linear spatiotemporal measures examined here [8,9]. The IMUs recorded 3D accelerations at 128 Hz and transmitted data to APDM Mobility Lab software on a Surface laptop (Microsoft, Redmond, WA, USA). Participants were instructed to perform four 1 min quiet standing trials under four sensory conditions: (1) standing on hard surface (H) with eyes open (EO), (2) standing on H with eyes closed (EC), (3) standing on a 48 × 40 × 6.2 cm thermoplastic elastomer foam (F) with EO, and (4) standing on F with EC. The 1 min length of each trial allowed calculations of jerk and range of acceleration to be made by the APDM Mobility Lab software, as well as obtaining above 6000 datapoints which is deemed reliable for sample entropy calculation during posturographic assessments [15].

Participants were allowed to perform trials shoed if wearing comfortable walking shoes, or barefoot if wearing unsafe shoes (e.g., high heels). The length of rest between trials was determined by each participant depending on their fatigue level after each trial. None of the participants identified fatigue as an issue in performing the trials. The participants stepped off the foam for at least 30 s between trials to avoid footprint depth marks on the foam. During trials, participants were instructed to stand with their arms at their sides and were spotted by a registered physiotherapist. The trials were all conducted in the same clinical space near a wall on the left side and facing the end of a ~10 m corridor that had a closed, windowed, double-leaf door with signs on it and above. To note, the environment did not present any moving elements that may have affected standing posture. Participants were not instructed to look at any specific section of the environment, but just to look forward. At the beginning of each test, a rhomboid-shaped block was placed between the feet of participants to ensure a consistent position across trials (between-heels distance of 10 cm, with toes turned out 10°). All participants performed all trials according to the instructions and none lost balance, required assistance, or evidently changed their posture during the trial.

***Data Analysis:*** 3D acceleration data was exported and processed in Matlab R2021a (Natwick, MA, USA). Acceleration in the anteroposterior (AP) and mediolateral (ML) directions from the sternum and lumbar sensors were used separately to calculate SE, utilising the equation by Richman and Moorman (2000) [16] using a tolerance of *r* = 0.2 (standard deviation) and a template size *m* = 2 [17] (Equation (1)). In the SE equation below, B(r)im is defined as (N – m − 1) times the number of vectors of length *m* that match a predetermined template, found by comparing their Chebyshev distance using the *r* tolerance value (excluding self-comparisons). This process is then repeated for *m* + 1 and *r* to produce $A^{m}\left(r\right)$. To note, for the SE calculations, we used the middle 54 s of data recording to avoid any effect of task transitions; hence, *N* = 6912 (128 Hz × 54 s).
(1)$$SE\;\left(m,\;r,\;N\right)=-ln\frac{A^{m}(r)}{B^{m}(r)}$$
where:Bmr=∑j=1N−mB(r)imN−m
and
Amr=∑j=1N−mA(r)imN−m

Four SE measures were obtained for each of the four tasks, EO-H, EO-F, EC-H, and EO-F, from each sensor. We also obtained two of the previously described linear measures that were identified from the literature to best classify pwMS: ML acceleration range during EO-F (AR_EO-F-ML_) and mediolateral jerk during EC-H (Jerk_EC-H-ML_). The latter two measures were obtained using the Mobility Lab software (APDM, Portland, OR, USA).

***Statistical Analysis:*** A multivariate ANOVA using groups (pwMS and HC) as well as subgroups (pwMS with an EDSS score below and above 2.0; pwMS^0–2^ and pwMS^2.5–4^, respectively) as fixed factors was used to determine between-groups differences for all metrics (16 SE measures (×2 sensors, ×2 directions, ×4 conditions) and two linear (Jerk and AR)). Significance was set at *p* < 0.05. Pairwise post-hoc comparisons were conducted for the subgroups analysis. A Bonferroni correction was applied to all comparisons.

A receiver operating characteristic (ROC) analysis was used to determine the sensitivity and specificity of all SE values and linear metrics (AR_EO-F-ML_ and Jerk_EC-H-ML_) to differentiate between HC and pwMS^0–4^. A secondary ROC analysis was conducted to determine classification performance for mild MS using only the data from 37 participants with an EDSS score ≤ 2.0 (pwMS^0–2^). The following classification performance based on the area under the curve (AUC) of the ROC analysis was adopted: AUC < 0.6 = fail, 0.6 ≤ AUC < 0.7 = poor, 0.7 ≤ AUC < 0.8 = fair, 0.8 ≤ AUC < 0.9 = good, and AUC > 0.9 = excellent [18]. A correlation coefficient was used to identify potential confounders such as age, height, and body mass index (BMI); no confounders were identified. All statistical analyses were conducted using IBM-SPSS 29 (Armonk, NY, USA).

## 3. Results

### 3.1. Participant Demographics

Demographics for the HC, pwMS^0–4^, and pwMS^0–2^ groups are presented in Table 1. Overall, pwMS (n = 58) presented with a median EDSS = 2.0. For pwMS^0–2^ (n = 37), the median EDSS value was 1.0. EDSS values were obtained from the latest score entered into MS-Base by a MS specialist neurologist [19]. No significant differences in age, height, or sex ratio between HC and pwMS^0–4^ or between HC and pwMS^0–2^ were found.

### 3.2. Between-Group Differences

pwMS^0–4^ had significantly greater SE_EC-H-AP_ (sternum and lumbar) than HC (Table 2). When on a foam surface, all SE values were significantly lower in pwMS, except SE_EO-F-ML_ and SE_EC-F-ML_ for the sternum. No significant between-group differences were found in SE when on a hard surface, except for SE_EC-H-AP_ (sternum and lumbar). No significant between-group differences were found for AR_EO-F-ML_ (*p* = 0.365) or Jerk_EC-H-ML_ (*p* = 0.585). Significant between-subgroups differences were found, mainly for tasks performed on foam (Figure 1). Significant between-subgroups differences were also found for SE_EC-H-AP_ (sternum and lumbar). Post-hoc analysis revealed that pwMS^0–2^ and pwMS^2.5–4^ SE values were significantly different for the lumbar sensor only during EC on foam in both AP and ML directions (*p* = 0.010 and *p* = 0.003, respectively).

### 3.3. Classification of pwMS^0–4^

Area under the ROC curve (AUC) values for all measures are presented in Table 2. Compared to the linear metrics, Jerk_EC-H-ML_ (AUC = 0.501, 95% CI [0.361–0.640]) and AR_EO-F-ML_ (AUC = 0.616, 95% CI [0.476–0.756]), lumbar SE_EC-H-AP_ demonstrated the best (fair) classification performance (AUC = 0.783, 95% CI [0.679–0.886]). The ROC curve (Figure 2a) shows that SE_EC-H-AP_ performed best overall up to a threshold of 80% true positive rate, with the false positive rate (FPR) remaining at 40%. At the same threshold, the other measures have a FPR over 60–70%.

### 3.4. Classification of pwMS^0–2^

Compared to Jerk_EC-H-ML_ (AUC = 0.470, 95% CI [0.316–0.624]) and AR_EO-F-ML_ (AUC = 0.569, 95% CI [0.418–0.720]), lumbar SE_EC-H-AP_ demonstrated a good classification performance (AUC = 0.810, 95% CI [0.700–0.920]) for milder disability (Figure 2b).

## 4. Discussion

This study was motivated by the need for more sensitive measures of balance to monitor disability at early stages of MS for the purpose of initiating timely interventions (e.g., physical rehabilitation and/or disease-modifying therapy changes). We calculated SE under all combinations of eye (EO and EC) and surface (H and F) conditions. In line with our hypothesis, we found that the best classifier was obtained from the lumbar sensor, during EC, when on a hard surface (H), and in the AP direction (SE_EC-H-AP_). Further, SE_EC-H-AP_ exhibited the best categorisation performance, even for pwMS^0–2^.

We found that most SE measures were significantly lower in pwMS than in HC during tasks on foam, signifying more regularity, but not when on a hard surface. Conversely, sternum and lumbar SE were significantly *higher* in pwMS than in HC on a hard surface with eyes closed (SE_EC-H-AP_), signifying less regularity in what ought to be a less complex task. Interestingly, the latter measures were also the best classifiers for both pwMS^0–4^ and pwMS^0–2^. Significant between-subgroups differences, except in the sternum ML, were found mostly during proprioceptively perturbing tasks (EO and EC on foam), where pwMS^2.5–4^ exhibited the lowest SE values. These findings are in line with Carpinella et al.’s (2022) study [13], which showed significantly lower SE values (AP and ML from a sternum placed sensor) in pwMS with low disability than in HC. Swanson et al.’s (2021) paper, which used linear metrics obtained from a lumbar placed sensor, also indicate that more complex tasks (EC-F) can highlight postural behaviour differences between healthy adults and pwMS [20]. The latter study also showed that some of the variables exhibited excellent discriminatory abilities; however, their sample of pwMS had a wider range of disability level (EDSS 0–6.5) compared to our sample of pwMS (EDSS 0–4). To note, EDSS values ≤ 4.5 indicate full ambulation capabilities [3].

Using approximate entropy (ApEn) on CoP data, Roeing et al. (2016) found that pwMS exhibited lower ApEn in the ML direction during EO on a hard surface than HC [14]. Similar findings were reported by Huisinga et al. (2012), who also found reduced ML ApEn in pwMS with eyes closed [21]. These finding are supported by Sun et al. (2019), who identified lower SE in the anteroposterior direction as the strongest feature to discriminate between pwMS with low risk of falling and HC [12]. Contrary to previous findings from the hard surface condition, we found no significant between-group differences in SE during EO. Furthermore, during EC, our findings are opposite to the EO condition for the lumbar and sternum SE (AP). The latter may be explained by differences in entropy calculation [7], SE parameter selection [22], data source (e.g., CoP vs. Acceleration), and/or disability levels (e.g., higher EDSS scores) of pwMS included in previous studies [12,14,21]. However, sensorimotor integration differences may also underlie our findings. Increased SE is thought to reflect greater complexity, improved self-organization of the sensorimotor system, and greater automaticity of balance responses [23]. On the other hand, decreased SE is often thought to indicate tighter balance control with greater cortical involvement, and hence more predicable motor responses, particularly during more challenging postural tasks [23,24]. Although the above interpretation of SE might explain our findings during postural tasks on a foam, where we found reduced SE during EC, it does not align with our SE_EC-H-AP_ findings (greater in pwMS than HC). Since SE_EC-H-AP_ values are similar to SE_EO-H-AP_ in pwMS, this may indicate two things: (a) that visual inputs are less relevant to maintain the complexity of postural responses, and/or (b) that pwMS at early stages may have adapted to compensate the absence of visual inputs by up-weighting proprioceptive and vestibular inputs. The latter may explain why SE during foam tasks is significantly lower in pwMS than in HC, as they may be more affected by the disturbance of proprioceptive inputs. This interpretation is in line with a previous study using frequency spectrum analysis of the CoP, which found lower power in the low frequencies (visual regulation of posture) in pwMS compared to HC [25].

An alternative explanation for our SE_EC-H-AP_ findings could be that pwMS may have perceived EC as a postural threat. The effects of postural threats using surface translations was investigated in a group of healthy adults, who showed an increased SE (AP) prior to perturbation [23]. The authors posit that postural threats may increase noise in the input to the balance control system, resulting in increased SE [23]. Increased SE (AP) has also been observed in older adults during EC, which is thought to reflect an inability to adequately re-weight sensory information, and/or precisely perceive and utilise the available inputs (e.g., proprioceptive, vestibular) due to inefficient cognitive/cortical involvement [26]. Previous studies have shown that corticospinal and neuromuscular changes at early stages of MS disease are associated with motor performance (i.e., increased muscle co-activation) and may also contribute to cortical control impairment [27,28,29]. Since cognitive-postural interference has been previously shown in pwMS [30], further studies should determine if it may also impact the automaticity of postural responses using SE.

We decided to compare discriminant abilities of SE measures with previously identified pwMS classifiers; however, it is noteworthy that the sensor location and task conditions and/or acceleration direction used are different [8,9]. For example, Spain et al. (2012) investigated jerk and used the same task conditions (EC-H), yet found differences in the ML direction, whereas Solomon et al. (2015) used EO-F (AR) also in the ML direction [8,9]. Better classification performance by sternum and lumbar SE_EC-H-AP_ in this study may be due to our pwMS sample being more homogeneous and having lower disability (EDSS) than in previous studies [8,9]. This may also suggest that SE is more sensitive to subtle postural impairments in MS.

SE is a measure of the regularity of the timeseries whereas jerk and AR measure overall smoothness and maximum amplitude of acceleration, respectively [8,9]. Increases in jerkiness as well as greater acceleration range may occur in more disabled pwMS, in whom slower sensorimotor responses due to proprioceptive impairments may cause acceleration changes of greater magnitude [31,32]. Unlike linear measures, which found differences in the ML direction, we found that SE was significantly greater in pwMS in the AP direction when standing on a hard surface with EC. During standing, postural control, occurring largely in the sagittal plane, is achieved by the ankle musculature, which has been previously shown to exhibit several physiological changes in pwMS [32]. Further, the EC condition demands an increased reliance on vestibular and proprioceptive sensory channels, both of which are also shown to be affected in pwMS [31,32]. However, and counterintuitively, the best classifier was lumbar SE_EC-H-AP_, i.e., not during more proprioceptively challenging tasks (foam surface). Again, MS-related sensorimotor integration issues described above may also explain this finding.

In this study we used IMU-derived measures of postural control, which differ from more traditional methods of posturography (i.e., CoP sway). However, the use of IMUs has seen rapid acceptance in clinical settings due to their portability and relative low cost when compared to, e.g., forceplates [33]. Although IMUs can provide abundant 3-dimensional data (e.g., acceleration and gyroscopes), we focussed here on obtaining linear measures from timeseries and postural tasks that have previously shown adequate discriminant performance. It is noteworthy that other non-linear measures may increase classification performance and/or complement current findings.

A better classification performance of SE over linear measures shows that this measure should be considered in the assessment and monitoring of progression, fall risk, and potentially as an adjunct to the complex diagnosis of MS. Further studies should also explore the responsiveness of SE to pharmacological and non-pharmacological interventions and its use as a clinical outcome measure. Since SE was obtained from acceleration data, standardization and automation of SE calculations would support its implementation using inertial sensors from mobile phones, which could help pwMS to self-monitor and potentially self-manage MS symptoms.

### Limitations

For the SE calculations we used recommended *m* = 2 and *r* = 0.2 values and a 1 min trial at 128 Hz [17]. Although the latter parameters’ values are predominant in the literature, changes in these values may lead to different results and should be carefully considered when comparing across studies [7]. The effect of *m* and *r* on SE have been previously discussed; however, the exact underlying biological processes that the selection of these parameters represent are not yet fully understood and should be further investigated [34]. Although we found fair to good pwMS classifiers using SE, other non-linear metrics such as multiscale sample entropy may offer a better performance.

Participants wore their own comfortable shoes during the assessment (e.g., no high heels), and we believe that this allows for a more ecologically valid approach [35]. Increased fatigue is common in pwMS and may influence postural stability and increase the risk of falling [11,36]. Although we did not objectively control for the latter, participants felt comfortable throughout the session, and none withdrew from the study. Dual tasks when assessing posture have been previously used to study pwMS, as cognition is often affected [30]. Assessing dual (motor-cognitive) postural tasks may increase classification performance when using SE and should be further examined.

## 5. Conclusions

We found lower SE in pwMS than HC when performing postural tasks on a foam, which indicates tighter postural control. However, sample entropy in the anteroposterior direction with eyes closed on a hard surface (SE_EC-H-AP_) was significantly higher in pwMS than in HC and was also the best classifier of pwMS even in those with mild MS (EDSS 0–2). SE may prove a useful tool to detect subtle MS progression and intervention effectiveness.

## Figures and Tables

**Figure 1 sensors-24-00872-f001:**
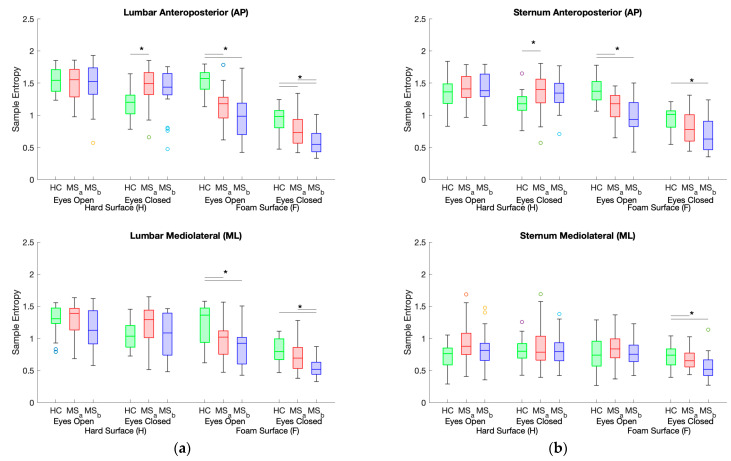
Boxplots present sample entropy (SE) values for both sensors (lumbar and sternum) and both directions (AP and ML) for all four conditions. HC = healthy controls, MS_a_ = pwMS with EDSS ≤ 2.0 (pwMS^0–2^), MS_b_ = pwMS with EDSS > 2.0 (pwMS^2.5–4^). * Horizontal lines indicate significant pairwise differences (post-hoc after Bonferroni correction, *p* < 0.005).

**Figure 2 sensors-24-00872-f002:**
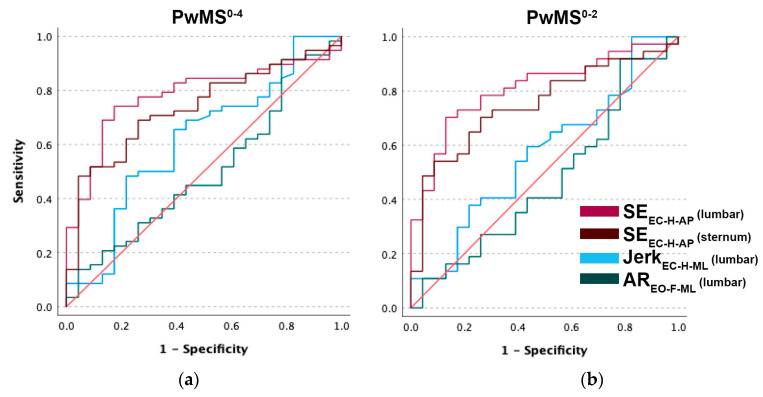
(**a**) ROC curves comparing the sensitivity and specificity of lumbar and sternum SE_EC-H-AP,_ AR_EO-F-ML_, and Jerk_EC-H-ML_ in pwMS^0–4^, and (**b**) in pwMS^0–2^.

**Table 1 sensors-24-00872-t001:** Participant characteristics.

Variable	HC	PwMS (EDSS 0–4.0)	PwMS (EDSS 0–2)
n	23	58	37
EDSS (median)	-	2.0	1.0
Years since diagnosis (mean ± SD)	-	6.0 ± 3.7	4.9 ± 3.2
Age (mean ± SD)	43.7 ± 12.0	42.9 ± 11.1	40.8 ± 10.7
Sex (male/female)	7/16	19/39	13/35
Height (cm) (mean ± SD)	172.0 ± 10.4	170.9 ± 9.6	172.1 ± 9.7
Weight (kg) (mean ± SD)	73.1 ± 14.0	81.3 ± 16.4	79.7 ± 16.5
Body mass index (kg/m^2^) (mean ± SD)	24.9 ± 4.6	27.8 ± 4.9	27.3 ± 5.1

EDSS, expanded disability status scale; HC, healthy controls; pwMS, people with multiple sclerosis.

**Table 2 sensors-24-00872-t002:** Descriptive, MANOVA, and AUC results for all metrics, conditions, and groups.

					HC		pwMS				
Metric	Eyes	Surface	Direction	Sensor	Mean	Sd	Mean	Sd	* p *	η^2^	AUC
SE	EO	H	ML	Lumbar	1.294	0.218	1.234	0.273	0.352	0.011	0.442
SE	EO	H	AP	Lumbar	1.539	0.176	1.545	0.278	0.932	0.000	0.542
SE	EO	H	ML	Sternum	0.737	0.208	0.902	0.300	0.018	0.069	0.648
SE	EO	H	AP	Sternum	1.349	0.238	1.415	0.236	0.262	0.016	0.577
SE	EC	H	ML	Lumbar	1.049	0.214	1.149	0.306	0.156	0.025	0.624
** *SE* **	** *EC* **	** *H* **	** *AP* **	** *Lumbar* **	** *1.183* **	** *0.230* **	** *1.431* **	** *0.302* **	** *0.001* **	0.138	** *0.783* **
SE	EC	H	ML	Sternum	0.795	0.197	0.848	0.291	0.419	0.008	0.529
** *SE* **	** *EC* **	** *H* **	** *AP* **	** *Sternum* **	** *1.178* **	** *0.193* **	** *1.370* **	** *0.270* **	** *0.003* **	0.109	** *0.736* **
*SE*	*EO*	*F*	*ML*	*Lumbar*	*1.222*	*0.311*	*0.949*	*0.288*	*<0.001*	0.152	*0.276*
*SE*	*EO*	*F*	*AP*	*Lumbar*	*1.534*	*0.193*	*1.070*	*0.298*	*<0.001*	0.376	*0.099*
SE	EO	F	ML	Sternum	0.754	0.257	0.820	0.206	0.227	0.018	0.577
*SE*	*EO*	*F*	*AP*	*Sternum*	*1.371*	*0.198*	*1.086*	*0.249*	*<0.001*	0.234	*0.190*
*SE*	*EC*	*F*	*ML*	*Lumbar*	*0.815*	*0.193*	*0.647*	*0.200*	*0.001*	0.131	*0.266*
*SE*	*EC*	*F*	*AP*	*Lumbar*	*0.924*	*0.225*	*0.702*	*0.233*	*<0.001*	0.162	*0.234*
SE	EC	F	ML	Sternum	0.709	0.168	0.629	0.181	0.072	0.040	0.361
*SE*	*EC*	*F*	*AP*	*Sternum*	*0.943*	*0.193*	*0.760*	*0.247*	*0.002*	0.113	*0.274*
Range	EO	F	ML	Lumbar	0.261	0.123	0.313	0.162	0.171	0.024	0.616
Jerk	EC	H	ML	Lumbar	2.602	3.347	3.292	5.644	0.585	0.004	0.501

AP, anteroposterior; AUC, area under the curve from receiver operating curve (ROC) analysis; EC, eyes closed; EO, eyes open; F, foam surface; H, hard surface; η^2^, partial eta squared. Significant differences are highlighted in italics (Bonferroni corrected, *p* < 0.005,). Highest AUC in bold.

## Data Availability

Data used and/or analysed in this study is available from the corresponding author on reasonable request.

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
