# Peer review of "Sample Entropy Improves Assessment of Postural Control in Early-Stage Multiple Sclerosis"

_sensors, 2024, doi:10.3390/s24030872_

Round 1
Reviewer 1 Report
Comments and Suggestions for Authors
In short, this manuscript aimed to assess whether sample entropy was more discriminative to postural instability when compared to two other acceleration measures (i.e., jerk and acceleration range) in people with MS having low disease severity score. The authors assessed participants under multiple stance conditions. This is a well written manuscript, though there are a variety of concerns that should be addressed, to productively augment the exciting literature.
Primary Comments:
Given only 10 participants had an EDSS ≥ 2.5, did those ten participants influence the results, or differ between the EDSS 0-2.0 group. If the authors decide to assess group differences, it would be appropriate to assess differences between the entire cohort, the EDSS 0-2.0 cohort, and the EDSS 2.5-4.0 cohort.
Items to enhance the protocol section:
- The protocol does not mention the EC firm condition, nor does it describe the duration for the foam trials.
- Where all trials 60 seconds? How was 60 seconds determined as the duration for trials?
- Did participants wear shoes or were they unshod?
- Did all participants complete the entire duration of all four conditions (that’s a relatively long duration to stand eyes closed on the foam surface), if not it should be described which conditions and how many participants did not complete the full trial and then make sure that is discussed in the analysis section.
- Did any participants touch the nearby wall, if so, was the trial ended or continued?
- Was there a fixation point for participants to look at during the eyes open trials, if so, how far away, if not what were the instructions?
- Were trials ended if eyes were opened during the eyes closed trials or moved their hands/arms from their side?
- Were participants given more than one opportunity if balance was lost per-condition?
- Was there a break between foam conditions to stand on the firm surface, or flip the foam pad?
Regarding stability measures, it is not clear why the authors decided to assess both the sternum and lumbar, what was the justification. Is there literature indicating one location would be better based on balance strategies? Was there a significant difference between sample entropy for those two locations? Why did the authors only assess Range and Jerk of one stance condition, given Mobility Lab provides data for each condition it would be prudent to report data from each condition. Also why did they only include those measures? There is a recent publication that might be worth reviewing, as the authors performed a similar discriminatory analysis on mobility (including static postural stability) comparing people with MS and healthy controls (https://doi.org/10.1016/j.msard.2021.102924).
For the analysis section, please provide greater detail. What type of multivariate ANOVA was used, what factors were included, did the authors perform a RMANOVA to assess differences between conditions and groups? It’s not entirely clear why the authors corrected for multiple comparisons given the measures and conditions were distinct, however, if there is a reason this reviewer is unaware of is the Bonferroni method too conservative of an approach? Please provide descriptive classification for AUC values (i.e., how should readers interpret an AUC=0.70).
The authors correctly highlighted the discrepancy between their results and prior results, then indicate that the differences could be due to their parameters used for quantifying SE. It may therefore be beneficial to include the formula used to calculate SE in the text, as well as any supporting literature discussing differences between different parameters and various outcomes.
Minor Comments:
Please describe the order of the subscripts and maintain consistency regarding the position of each descriptor (e.g., surface, visual condition, direction). Also, there are instances where there is a superscript rather and subscript (line 85).
Line 39 – there should be an ‘an’ between “and EDSS”
Line 48 – there is a space between two of the citations, they should be combined within the same bracket.
For Table 1: provide numbers for sex distribution (10m, 30f), add average disease duration, add weight. Some of the lines are italicized but not bold, it would be helpful to add a line between stance conditions.
How was the EDSS assessed/quantified for each participant?
Author Response
Response to Reviewers
First of all, we would like to thank the reviewers for their insightful and constructive feedback, which has with no doubt helped to improve the quality of the manuscript. Below we have responded to all comments on a point-by-point basis and signed where changes to manuscript were made. We have also updated the literature according to the suggested readings and more recent literature.
Reviewer #1
In short, this manuscript aimed to assess whether sample entropy was more discriminative to postural instability when compared to two other acceleration measures (i.e., jerk and acceleration range) in people with MS having low disease severity score. The authors assessed participants under multiple stance conditions. This is a well written manuscript, though there are a variety of concerns that should be addressed, to productively augment the exciting literature.
We appreciate the reviewer’s positive feedback.
Primary Comments:
- Given only 10 participants had an EDSS ≥ 2.5, did those ten participants influence the results, or differ between the EDSS 0-2.0 group. If the authors decide to assess group differences, it would be appropriate to assess differences between the entire cohort, the EDSS 0-2.0 cohort, and the EDSS 2.5-4.0 cohort.
We are very grateful about this comment as it highlighted a misleading typo issue regarding the n for pwMS0-2. For this subgroup n is 37 and represents 64% of our sample. Also, we did consider the reviewers recommendation about subgroup comparisons, which we have added in the methods sections, added to Figure 1, and discussed accordingly. Overall, the results of the pairwise post-hoc comparisons (Bonferroni corrected) show significant differences between HC and the 2 pwMS groups (pwMS0-2 and pwMS2.5-4) for most tasks when on foam. The other 2 significant differences are in line with our AUC-ROC classification, which indicates that SEEC-H-AP exhibited the best classification performance for the whole pwMS group and for those with mild disability (pwMS0-2). Please refer to tracked manuscript section 3.2. We have also added the following text to the discussion (paragraph 2 below in response to Q 13).
Items to enhance the protocol section:
The following questions (2-10) have been all addressed in the “Protocol” subsection for which the text below has been added/modified:
- The protocol does not mention the EC firm condition, nor does it describe the duration for the foam trials.
- Where all trials 60 seconds? How was 60 seconds determined as the duration for trials?
- Did participants wear shoes or were they unshod?
- Did all participants complete the entire duration of all four conditions (that’s a relatively long duration to stand eyes closed on the foam surface), if not it should be described which conditions and how many participants did not complete the full trial and then make sure that is discussed in the analysis section.
- Did any participants touch the nearby wall, if so, was the trial ended or continued?
- Was there a fixation point for participants to look at during the eyes open trials, if so, how far away, if not what were the instructions?
- Were trials ended if eyes were opened during the eyes closed trials or moved their hands/arms from their side?
- Were participants given more than one opportunity if balance was lost per-condition?
- Was there a break between foam conditions to stand on the firm surface, or flip the foam pad?
“Protocol: Postural control was measured using two Opal IMUs (APDM, Portland, OR, USA) placed at the sternum and lumbar spine. We selected both locations as previous studies have identified pwMS classifiers using metrics from both sites, using the linear spatiotemporal measures examined here [8,9]. IMUs recorded 3D accelerations at 128 Hz and transmitted data to the APDM Mobility Lab software on a Surface laptop (Microsoft, Redmond, WA, USA). Participants were instructed to perform four 1-minute quiet standing trials under four sensory conditions: 1) standing on hard surface (H) with eyes open (EO), 2) standing on H with eyes closed (EC), 3) standing on a 48x40x6.2 cm thermoplastic elastomer foam (F) with EO, and, 4) standing on a foam with EC. The 1-minute length of each trial allowed the calculations of jerk and range of acceleration by` the APDM Mobility Lab software as well as obtaining above 6000 datapoints which is deemed reliable for sample entropy calculation during posturographic assessments [14].
Participants were allowed to perform trials shoed if wearing comfortable walking shoes and barefoot if wearing unsafe shoes (e.g., high heels). The length of the rest between trials was determined by each participant depending in their fatigue level after each trial. None of the participants identified fatigue as an issue to perform the trials. The participants stepped off the foam for at least 30 seconds between trials to avoid footprint depth marks on the foam. During a trial participant were instructed to stand with their arms at their sides and were spotted by a registered physiotherapist. The trials were all conducted in the same clinical space near a wall on the left side and facing the end of a ~10m corridor that had a closed windowed double-leaf door with signs on it and above. To note, the environment did not present any moving elements that may have affected standing posture. Participants were not instructed to look at any specific section of the environment but just look forward. At the beginning of each test, a rhomboid-shaped block was placed between the participant's feet to ensure a consistent position across trials (between-heels distance of 10 cm with toes turned out 10°). All participants performed all trials according to the instructions and none lost balance, required assistance, or evidently changed their posture during the trial.”
- Regarding stability measures, it is not clear why the authors decided to assess both the sternum and lumbar, what was the justification. Is there literature indicating one location would be better based on balance strategies? Was there a significant difference between sample entropy for those two locations?
Although interesting per se, comparing the SE results from both locations is beyond the aims of this paper. The following text has been added to explain the selection of both locations for sensors’ placement:
“We selected both locations as previous studies have identified pwMS classifiers using metrics from both sites using linear spatiotemporal measures examined here[8,9]”.
- Why did the authors only assess Range and Jerk of one stance condition, given Mobility Lab provides data for each condition it would be prudent to report data from each condition. Also why did they only include those measures?
Mobility Lab indeed produces a large number of measures for each condition. We felt that analysing all measures, without a hypothesis-driven question, would increase the risk of a type I error. We chose to build on finding from evidence to date. However, we agree that the reason for selecting these measures was not properly highlighted in the introduction. The following text have been added to paragraph 3 and 5, respectively:
“Linear measures of postural control are most commonly used to quantify balance impairments with IMUs in clinical populations. Although IMU-based systems can provide a broad range of linear measures, few of them have shown to be able to discriminate and/or differentiate between healthy controls (HC) and pwMS with mild-to-low disability, namely: range of acceleration,[8] and jerk.[9].”
“Hence, it is not known whether non-linear posturography measures, e.g. SE, may better identify subtle impairments of motor control in pwMS with milder symptoms than currently identified linear measures (e.g., jerk and range of acceleration).”
- There is a recent publication that might be worth reviewing, as the authors performed a similar discriminatory analysis on mobility (including static postural stability) comparing people with MS and healthy controls (https://doi.org/10.1016/j.msard.2021.102924)
Thanks for recommending this paper. We have added and discussed accordingly. The following paragraph has been modified to include the presented information:
“We found that most SE measures were significantly lower in pwMS than in HC during tasks on foam, signifying more regularity, but not when on a hard surface. Conversely, sternum and lumbar SE were significantly higher in pwMS than in HC on a hard surface with eyes closed (SEEC-H-AP), signifying less regularity in what ought to be a less complex task. Interestingly, the latter measures were also the best classifiers for both pwMS0-4 and pwMS0-2. Significant between subgroups differences were found mostly (except sternum ML) during proprioceptively perturbing tasks (EO and EC on foam), were pwMS2.5-4 exhibited the lowest SE values. These findings are in line with Carpinella et al.’s (2022) study, which showed significantly lower SE values (AP and ML from a sternum placed sensor) in pwMS with low disability than in HC. Swanson et al.’s (2021) paper, which used linear metrics obtained from a lumbar placed sensor, also indicate that more complex tasks (EC-F) can highlight postural behaviour differences between healthy adults and pwMS. [20] The latter study also showed that some of the variables exhibited excellent discriminatory abilities, however, their sample of pwMS had wide range of disability level (EDSS 0-6.5) compared to our sample of pwMS (EDSS 0-4). To note, EDSS values ≤ 4.5 indicate full ambulation capabilities [3].”
- For the analysis section, please provide greater detail. What type of multivariate ANOVA was used, what factors were included, did the authors perform a RMANOVA to assess differences between conditions and groups? It’s not entirely clear why the authors corrected for multiple comparisons given the measures and conditions were distinct, however, if there is a reason this reviewer is unaware of is the Bonferroni method too conservative of an approach? Please provide descriptive classification for AUC values (i.e., how should readers interpret an AUC=0.70).
For the MANOVA we used all SE measures obtained from the 2 sensors (x2) for the 4 conditions (x4) and for the 2 (ML and AP) directions (x2) plus the 2 linear metrics (jerk and range of acceleration). In total 18 dependant variables were used (16 SE measures + 2 linear) with disease group (HC or pwMS) as fixed factor. For the subgroup analysis we used 3 disease groups (HC, pwMS0-2, and pwMS2.5-4) as requested by this reviewer. All analyses (including post-hoc) used Bonferroni corrections to decrease the risk of a type I error when making multiple statistical tests. The following text has been added to the Statistical Analysis subsection, which also includes AUC interpretation.
“Statistical Analysis: A multivariate ANOVA with group (pwMS and HC) as well as subgroups (pwMS with an EDSS score below and above 2.0; pwMS0-2 and pwMS2.5-4, respectively) as fixed factors was used to determine between-groups differences for all metrics (16 SE measures [x2 sensors, x2 directions, x4 conditions] and 2 linear [Jerk and AR]). Significance set at p<0.05. Pairwise posthoc comparisons were conducted for the subgroups analysis. A Bonferroni correction was applied to all comparisons.
A receiver operating characteristic (ROC) analysis was used to determine the sensitivity and specificity of all SE values and linear metrics (AREO-F-ML and JerkEC-H-ML) to differentiate between HC and pwMS0-4. A secondary ROC analysis was conducted to determine classification performance for mild MS using only the data from 37 participants with an EDSS score ≤2.0 (pwMS0-2). The following classification performance based on the area under the curve (AUC) of the ROC analysis was adopted: AUC < 0.6 = fail, 0.7 ≤ AUC < 0.8 = fair, 0.8 ≤ AUC < 0.9 = good, and AUC > 0.9 = excellent.[19] A correlation coefficient was used to identify potential confounders such as age, height, and body mass index (BMI); no confounders were identified. All statistical analyses were conducted using IBM-SPSS 29 (Armonk, NY, USA).”
- The authors correctly highlighted the discrepancy between their results and prior results, then indicate that the differences could be due to their parameters used for quantifying SE. It may therefore be beneficial to include the formula used to calculate SE in the text, as well as any supporting literature discussing differences between different parameters and various outcomes.
We have added the reference and equation for the SE calculation to the Data Analysis subsection. We have also added the following text to the discussion section where selection parameters are discussed:
“For the SE calculations we used recommended m = 2 and r = 0.2 values and a 1-minute trial at 128Hz. [18] Although the latter parameters’ values are predominant in the literature, changes in these values may led to different results and should be carefully considered when comparing across studies. [34] The effect of m and r on SE have been previously discussed, however, the exact underlying biological processes that the selection of these parameters represent, are not yet fully understood and should be further investigated. [35] Although we found fair to good pwMS classifiers using SE, other non-linear metrics such as multiscale sample entropy may offer a better performance.”
Minor Comments:
- Please describe the order of the subscripts and maintain consistency regarding the position of each descriptor (e.g., surface, visual condition, direction). Also, there are instances where there is a superscript rather and subscript (line 85).
Amended throughout the manuscript including table 2. We have adopted a “visual condition – surface – direction” for all acronyms in the manuscript.
- Line 39 – there should be an ‘an’ between “and EDSS”
Done
- Line 48 – there is a space between two of the citations, they should be combined within the same bracket.
Done
- For Table 1: provide numbers for sex distribution (10m, 30f), add average disease duration, add weight. Some of the lines are italicized but not bold, it would be helpful to add a line between stance conditions.
Requested information has been added/modified in table 1. For table 2, and as indicated in at the bottom of the table “Significant differences are highlighted in italics (Bonferroni corrected, p<0.005,). Highest AUC in bold”
- How was the EDSS assessed/quantified for each participant?
The EDSS was obtained from the clinical record that was the closest to the balance assessment time and confirmed by the treating neurologist. The following text has been added in the section 3.1: “EDSS values were obtained from the latest score entered into MS-Base by a MS specialist neurologist. [17]”

Reviewer 2 Report
Comments and Suggestions for Authors
The work was carried out at a good methodological level.
The results are beyond doubt thanks to well-conducted statistical analysis.
The discussion of the results is interesting.
If the authors would comment on the achievements and novelty of their study compared to Gomez-Hernandez et al. 2023. doi: 10.4103/jmss.jmss_184_21, this will improve the validity and usability of their results.
too much abbreviation makes the abstract difficult to read
Author Response
Response to Reviewers
First of all, we would like to thank the reviewers for their insightful and constructive feedback, which has with no doubt helped to improve the quality of the manuscript. Below we have responded to all comments on a point-by-point basis and signed where changes to manuscript were made. We have also updated the literature according to the suggested readings and more recent literature.
Reviewer #2
- The work was carried out at a good methodological level.
We appreciate the reviewer positive comments.
- The results are beyond doubt thanks to well-conducted statistical analysis.
We appreciate the reviewer’s positive feedback.
- The discussion of the results is interesting.
Thanks for the compliment on our work.
- If the authors would comment on the achievements and novelty of their study compared to Gomez-Hernandez et al. 2023. doi: 10.4103/jmss.jmss_184_21, this will improve the validity and usability of their results.
The text below has been added in the Introduction (par. 4) and citation (13) has been added in the Limitations section where fatigue is discussed:
“SE reduction has also been observed in pwMS who exhibit lower limb muscle fatigue [13]”
- Too much abbreviation makes the abstract difficult to read.
In the revised version of the manuscript, we have minimised the number of acronyms yet maintaining a maximum of 200 words

Round 2
Reviewer 1 Report
Comments and Suggestions for Authors
I would like to commend the authors on their revised manuscript, which is substantially improved. I have only minor comments for the authors.
- Throughout the manuscript intext citations are inconsistently placed around the period (e.g., “.[xx]” or “[xx].”).
- Lines 172 – 174: Given this reviewer assessed the provided citation, they believe the syntax for the AUC classification range is incorrect. For example, shouldn’t the range read that an AUC of greater than or equal to (≥) 0.6 and less than (<) 0.7 indicates poor classification? If correct the classification should be AUC < 0.6 = fail, 0.6 ≥ AUC < 0.7 = poor, 0.7 ≥ AUC < 0.8 = fair, 0.8 ≥ AUC < 0.9 = good, AUC > 0.9 = excellent
Author Response
Dear Editor, in the attached revised manuscript we have addressed Reviewer's #1 minor concerns. As noted and suggested, we have corrected AUC-ROC classification performance ranges which now read as follows: "AUC < 0.6 = fail, 0.6≤AUC < 0.7 = poor, 0.7≤AUC < 0.8 = fair, 0.8≤AUC < 0.9 = good, and AUC > 0.9 = excellent"
We have also thoroughly checked the entire manuscript for inconsistencies regarding in-text citations format and changed where it corresponded.
